# Evaluating Multi-Agent Coordination Abilities in Large Language Models

## Abstract

A pivotal aim in contemporary AI research is to develop agents proficient in multi-agent coordination, enabling effective collaboration with both humans and other systems. Large Language Models (LLMs), with their notable ability to understand, generate, and interpret language in a human-like manner, stand out as promising candidates for the development of such agents. In this study, we build and assess the effectiveness of agents crafted using LLMs in various coordination scenarios. We introduce the LLM-Coordination (LLM-Co) Framework, specifically designed to enable LLMs to play coordination games. With the LLM-Co framework, we conduct our evaluation with three game environments and organize the evaluation into five aspects: Theory of Mind, Situated Reasoning, Sustained Coordination, Robustness to Partners, and Explicit Assistance. First, the evaluation of the Theory of Mind and Situated Reasoning reveals the capabilities of LLM to infer the partner's intention and reason actions accordingly. Then, the evaluation around Sustained Coordination and Robustness to Partners further showcases the ability of LLMs to coordinate with an unknown partner in complex long-horizon tasks, outperforming Reinforcement Learning baselines. Lastly, to test Explicit Assistance, which refers to the ability of an agent to offer help proactively, we introduce two novel layouts into the Overcooked-AI benchmark, examining if agents can prioritize helping their partners, sacrificing time that could have been spent on their tasks. This research underscores the promising capabilities of LLMs in sophisticated coordination environments and reveals the potential of LLMs in building strong real-world agents for multi-agent coordination.

## 1 Introduction

Humans engage in various coordination tasks in their daily lives and work, including mundane activities like cooking and more important tasks like search and rescue. In order to assist humans with tedious or hazardous tasks, it is essential to create agents capable of coordinating with humans or other autonomous systems. Recently, agents based on Large Language Models have successfully demonstrated emergent problem-solving and task-completion capabilities in complex environments Raman et al. (2022); Wang et al. (2023); Wu et al. (2023). They have shown high-level reasoning abilities and hints of Theory of Mind abilities Kosinski (2023). In this work, we intend to find out how well Large Language Models can reason to solve tasks that require multi-agent coordination. Effective coordination requires agents to be able to infer their partner's next actions (**Theory of Mind**), reason about the inferred action in the context of their shared environment (**Situated Reasoning**), take actions and make adjustments to execute the plan over a long duration (**Sustained Coordination**) and be able to adjust to unseen partners (**Robustness to Partners**). Furthermore, we need agents to be capable of proactively providing **Explicit Assistance** to their partners during coordination tasks.

In order to evaluate the multi-agent coordination abilities of LLMs, we adopt three different coordination games. The first game is Collab Escape, where two agents need to coordinate to escape from an adversary. The second is Collab Capture where two agents chase an adversary through a maze of rooms. The final game is the Overcooked Carroll et al. (2019a), which requires two players to cook and deliver onion soups. To enable Large Language Models to understand and play these games we introduce the LLM-Coordination framework. The LLM-Co framework provides agents

with contextual state information and feasible actions and interprets agent outputs for execution in real time. We will refer to agents using the LLM-Coordination framework as LLM-Co agents.

In the evaluation, we first test the Theory of Mind (ToM) and Situated Reasoning abilities of LLMs, which are preliminary skills required for coordination. ToM allows models to infer the intentions and beliefs of others, while Situated Reasoning enables them to anchor these inferences in the contextual setting of the environment. We design the **LLM-ToM-Reasoning Test Set**, including independent scenarios from our multi-agent coordination environments. The LLM-ToM-Reasoning Test Set requires the LLMs to reason about their partner's intention and the current state of the environment to provide the optimal next action. We compare four different LLMs (GPT-4, GPT-3.5-turbo, Vicuna-33B, and Vicuna-13B) OpenAI (2023); Ouyang et al. (2022); Chiang et al. (2023). We observe that GPT-4 overwhelmingly outperforms the other LLMs, getting an almost human-level score.

In order to evaluate sustained coordination abilities in LLM-Co agents, we use GPT-4 as the LLM of choice as it is the only candidate that provides acceptable ToM and Situated Reasoning skills. We compare the performance of LLM-Co Agents (w. GPT-4) with Reinforcement Learning (RL) based baselines, which are the gold standards for AI-AI gameplay. We also experiment with varying the partners in the Coordination Environment to proxy human agents to test the agent's Robustness to Partners. We observe that LLM-Co agents perform better than or equal to the RL baseline in both AI-AI and AI-human proxy gameplay without any fine-tuning. Additionally, LLM agents have a further edge over RL methods due to their ability to fully explain the rationale behind their actions in free text.

Finally, we study whether LLM-Co agents can proactively provide help to their partner (Explicit Assistance). We extend the existing layouts in the Overcooked-AI environment to involve a gate element that forces agents to assist their partners in order to complete deliveries. Through experiments on these new layouts, we discover that LLM-Co agents can determine the right strategy needed to help out their partners. However, they require a "helper directive," which uses natural language to prompt the LLM to be attentive to situations where their partner may need such help. We show that LLM-Co agents are able to outperform MARL baselines on these new layouts as well.

We summarize the key contributions of our work as follows:

- We develop the LLM-Coordination Framework that equips Large Language Models with tools and contextual information allowing them to play long-horizon games and execute LLM-generated natural language actions in real-time.

- We present the LLM-ToM-Reasoning test set which consists of scenarios from the three coordination games explicitly designed to test the Theory of Mind and Situated Reasoning abilities of Large Language Models.

- Using GPT-4 (which performs best on the LLM-ToM-Reasoning test) as the LLM of choice, we perform evaluations for assessing sustained coordination. We show that LLM-Co agents outperform Reinforcement Learning baselines in comprehensive evaluations in the multi-turn Overcooked-AI environment.

- We introduce two new layouts to the Overcooked-AI environment that require Large Language Models to provide Explicit Assistance to their partners. Through quantitative and qualitative evaluations, we show that LLM-Co Agents understand the common-payoff nature of the game and are able to figure out the right actions and reasoning required to assist their partners.

## 2 RELATED WORK

### 2.1 MULTI-AGENT COORDINATION

In Game Theory, Pure Coordination games are situations where the payoff is commonly shared between both agents. In such situations, cooperating is the best strategy. Various benchmarks have been used to evaluate Multi-Agent Coordination abilities over the years Lowe et al. (2017); Bard et al. (2020). In recent years, the Overcooked environment has emerged as a popular testbed for coordination experiments Carroll et al. (2019a); Wu et al. (2021). Our research leverages the Overcooked-AI environment Carroll et al. (2019b). The foundational work by Carroll et al. (2019a)

emphasized the significance of incorporating human data for effective collaboration. Subsequent research has pivoted towards enabling self-play-trained agents to coordinate seamlessly with humans within this environment. These studies employ various techniques, including self-play with past agent checkpoints Strouse et al. (2021), centralized population entropy objectives Zhao et al. (2023), open-ended objectives using graph theory Li et al. (2023), policy ensembles with context-aware mechanisms Lou et al. (2023), and the incorporation of human biases as linear hidden rewards Yu et al. (2023), to enhance the training and diversity of AI agents in different scenarios. Embodied environments usually set up in household environments have also been recently used to study multi-agent collaboration Puig et al. (2021); Jain et al. (2020; 2019); Gan et al. (2021).

## 2.2 PLANNING AND REASONING WITH LARGE LANGUAGE MODELS

Large Language Models (LLMs) have demonstrated remarkable capabilities of reasoning in natural language OpenAI (2023); Ouyang et al. (2022); Chiang et al. (2023). These models have achieved state-of-the-art performance across a spectrum of NLP tasks, showcasing their proficiency at verbal reasoning. Strategies like Chain of thought promptingWei et al. (2022), which generates step-by-step free-text explanations before coming to conclusions have further boosted the reasoning capacities of LLMs. Approaches augmenting an LLM with memory, belief, and tools have shown to be useful in multi-step problem-solving Park et al. (2023); Huang et al. (2022); Raman et al. (2022). Isolated LLM agents have shown to be capable of life-long learning and task completion in open-domain survival games, outperforming existing SOTA Reinforcement Learning methods Wu et al. (2023); Wang et al. (2023). More recently, such LLM agents have been paired with rule-based low-level planners to execute tasks in embodied environments Liang et al. (2022); Song et al. (2022). Zhang et al. (2023) demonstrated efficiency increase in collaborative embodied multi-agent setting and Mandi et al. (2023) have shown the ability of collaborative manipulator motion planning using LLMs. Taking the planning and reasoning abilities of LLM a step further, We intend to perform a systematic evaluation of coordination abilities in Large Language Models in Common Payoff games.

## 3 EVALUATION ENVIRONMENTS

## 3.1 COLLAB CAPTURE

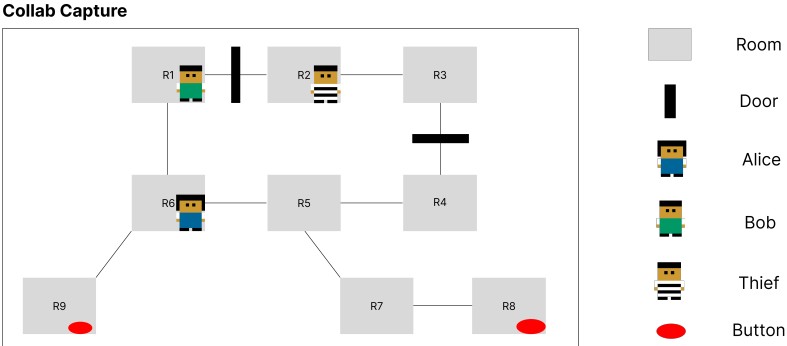

Figure 1: The CollabCapture game involves two agents, Alice (Blue) and Bob (Green), chasing a thief across multiple rooms. Some rooms are connected by doors, which can be controlled by buttons in different rooms.

Collab Capture involves two agents trying to capture an adversary in a maze of interconnected rooms. The rooms are connected by doors, which can be controlled through access buttons that can be found in different rooms. The agent's task is to capture the adversary in the least amount of time using effective strategies including cornering the adversary, disabling the adversary, or enabling their partners.

### 3.2 COLLABESCAPE

Based on the popular Video Game "Dead-by-Daylight", Collaborative Escape involves two agents trying to escape an adversary in a maze of interconnected rooms. They need to fix two generators located in randomly selected rooms to open an exit portal. The adversary tries to catch the agents, and the win condition is any one agent escaping. This game requires strategies like luring the adversary away from the partner, sacrificing for the partner's safety, and manipulating the movement of the adversary.

### 3.3 OVERCOOKED

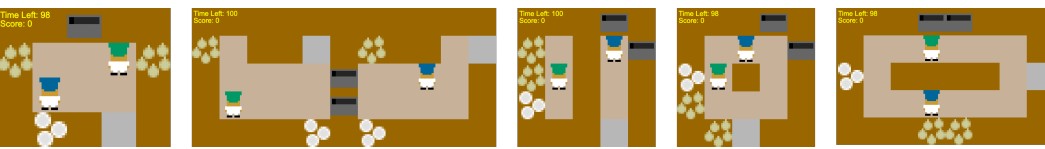

Figure 2: All layouts from the overcooked environment we use for our tests. The two agents Alice (Blue) and Bob (Green) need to collaborate to cook, plate, and deliver onion soups. From Left to Right: Cramped Room, Asymmetric Advantages, Forced Coordination, Coordination Ring, and Counter Circuit.

In the Overcooked-AI environment Carroll et al. (2019a), two agents—Alice (Blue) and Bob (Green)—collaborate to cook and deliver onion soups. Different environments feature varying numbers of onion dispensers ($o$), plate dispensers ($p$), cookers ($c$), delivery areas ($d$), and counters ($k$). Agents must load three onions into a cooker to start it, which takes 20 time steps to cook. Once done, an agent transfers the soup to a plate and delivers it.

### 3.4 OVERCOOKED-ASSIST: DEMONSTRATING EXPLICIT ASSISTANCE IN OVERCOOKED

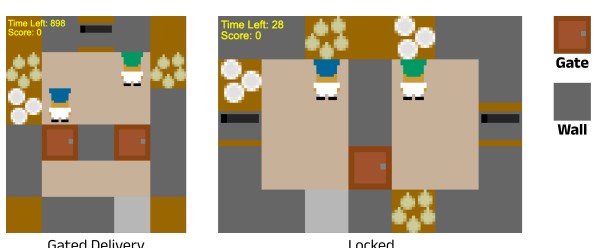

Figure 3: Additional Layouts that require agents to explicitly help their partner complete a delivery. These new layouts utilize **walls** and **gates** to create situations requiring explicit assistance.

The layouts in Overcooked-AI Carroll et al. (2019a) are an excellent test for gauging the ability of participating agents to sync their actions with their partners. It requires agents to effectively navigate the layout and time their actions in response to their partners in order to increase efficiency. However, none of these environments elicit the need for agents to explicitly help out their partner sacrificing their own time.

We intend to evaluate the LLM agent's ability to make the choice to actively help its partner but also see if they can realize when the opportunity to help has arrived and take the right action to facilitate their partner. If the LLMs cannot make such a choice implicitly, we intend to see the effect of tuning the directives to bring about such a cooperative intention. To elicit situations that require one agent to drop their own cooking/delivery and help out their partners, we extend the Overcooked environment by introducing 2 new facilities (Gates, Walls) and 2 new layouts.

### 3.4.1 GATED DELIVERY

Visualized in Figure 3, the Gated Delivery layout requires both agents to help out their partners during soup delivery. The two gates g0 and g1 make the delivery area inaccessible. Gates can be opened by an agent provided they are not holding anything in their hand. Once opened, a gate remains open for a very short time enough for an agent to move through it but not enough for an agent to open it in advance before picking up cooked soup for delivery. This necessitates an agent not holding cooked soup in their hand to go and open the gate for the delivery agent. The kitchen counters are replaced by walls to prevent the agents from taking the loophole of placing their soups temporarily on counters to open the gates. In this environment, both agents are equally placed, and they need to be acutely aware of their partner's needs in order to complete even a single delivery.

### 3.4.2 LOCKED

Visualized in Figure 3, the Locked environment is structurally similar to Soup Passing, except there is no shared counter to pass soup on. Instead, the agent in the left partition has to understand that their partner is locked behind a closed gate, holding a soup. In real-life scenarios, one collaborating partner might find themselves disadvantaged in a similar manner. In order to develop reliable assistive agents, the advantaged agent needs to understand the situation and make the choice to help out their partner since that provides a better common payoff.

## 4 LLM-COORDINATION FRAMEWORK

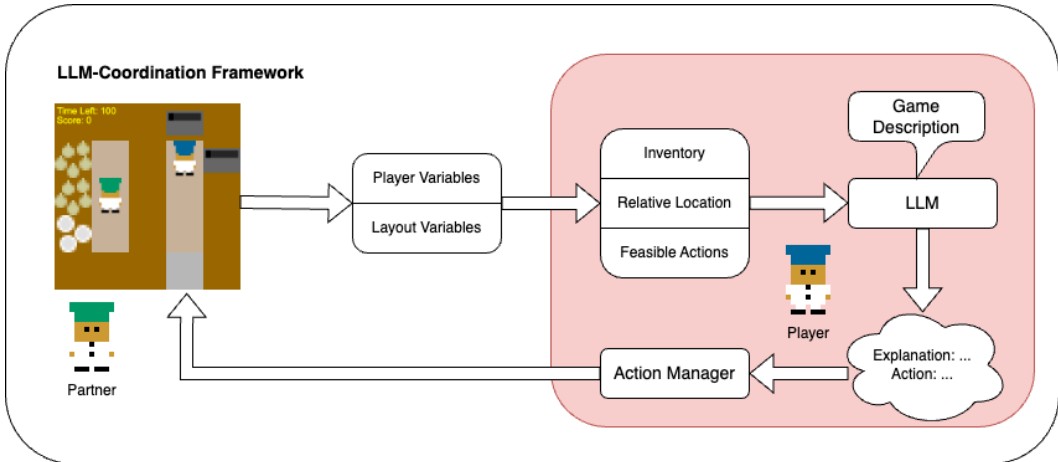

Figure 4: Visual summary of the LLM-Co framework. Our framework serves as the backbone for an individual agent, focusing on bringing out its coordination ability. The framework translates abstract game details into an LLM-compatible format and then utilizes the generated LLM output to take actions in the game world.

The games mentioned in Section 3 are translated into textual objectives using the LLM-Coordination Framework. The details of the game along with the rules and the layout of the map are condensed into a short **Game Description** ($G$). Along with the game description, we also provide a set of **Directives** ($D_i$) that guide agent behavior. These descriptions are passed as initial prompts to the Large Language Model.

At each turn, the LLM receives the current **state description** ($D(S)$) that is programmatically obtained from the environment, and the player states $S$. Since LLMs struggle with grid-based reasoning and navigation, we provide relative distances from the agent to each location of interest in the state description. Along with player-specific variables, other salient state variables are also included as natural language descriptions. Finally, an agent is provided its partner's inventory and relative position to allow it to consider their intentions. The state information provided to the LLM is equivalent to what a Reinforcement Learning agent would receive in the form of vectors.

The LLM operates at a **medium-level action space** which is made up of **verb-based actions** like "pick", "place", "move" etc. It is provided with a set of **feasible actions** $M_f$ to choose from to enable easier reasoning. The feasible action set is decided on the basis of player inventory and accessibility of locations.

The LLM utilizes the information $\langle G, D_i, S, M_f \rangle$ to assess the situation and generates an action $m$ from the provided set $M_f$. We then use an **Action Manager** to interpret the action based on the verb used and the location mentioned. The Action Manager generates low-level actions needed to execute the medium-level action. In the following experiments, we will refer to LLM Agents that use the LLM-Coordination Framework as **LLM-Co Agents**.

## 5 EXPERIMENTS AND RESULTS

In this section, we describe the experiments and results for the coordination ability of LLMs, with a focus on five aspects: Theory Of Mind, Situated Reasoning, Sustained Coordination, Robustness to Partners, and Explicit Assistance.

### 5.1 THEORY OF MIND AND SITUATED REASONING

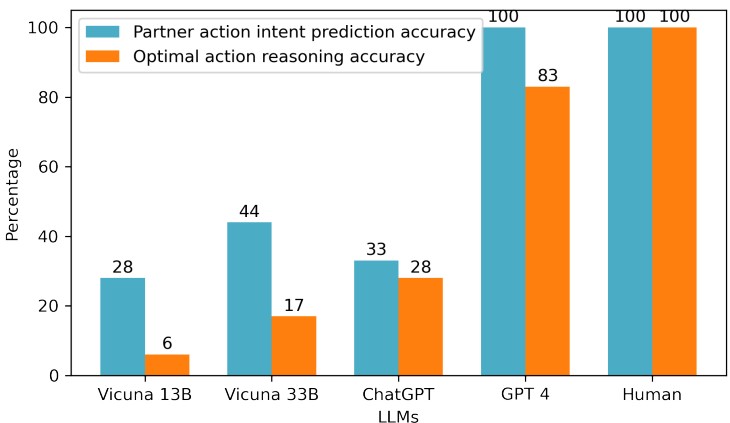

Figure 5: LLMs performance on the LLM-ToM-Reasoning test set. Partner action intent prediction accuracy shows the Theory Of Mind ability of LLMs under test and the optimal action reasoning accuracy infers the Situated Reasoning effect of LLMs under test. GPT-4 achieves the best performance among tested LLMs.

**LLM-ToM-Reasoning test set** With the LLM-Co frameworks, we propose an LLM-ToM-Reasoning test set, which is a suite of 18 scenarios posed with questions among all three games: Collaborative Capture, Collaborative Escape, and Overcooked. The scenarios in pure-text are formed by the outputs of the State Description and Feasible Action Generator from the LLM-Co frameworks. The test set only includes scenarios hand-picked to represent pivotal situations that require the agent under-test to first take their partner's possible next actions into active consideration, reason about the current state, and adjust their actions that "indirectly" lead to the best possible outcome. The same questions are shared across the test set which asks to analyze the current state, infer the partner's potential next action, and predict the optimal next action from the perspective of a player. We annotate the ground truth answers to the questions in the test set manually and ensure a 100% human success rate during the cross-validation process.

**GPT-4 outperforms the other LLMs in LLM-ToM-Reasoning test set** We use the collected LLM-TOM-Reasoning test set to scrutinize the LLMs in the Theory Of Mind (ToM) and Situated Reasoning aspects, which respectively refer to the ability to understand the beliefs and intentions of other entities and the ability to contextualize this understanding within the environmental dynamics

to formulate appropriate responses. LLMs under-test are required to solve the questions in the LLM-TOM-Reasoning test set, where the output answers to questions are manually compared against the ground truth. First, we calculate the accuracy of the LLM predictions concerning their partner's actions to indicate the ToM ability. Then, the accuracy of predictions for the next appropriate action and analysis of the current scenario shows the situated reasoning effectiveness. As the result shown in Figure 5, GPT-4 outperforms the other LLMs, GPT-3.5-turbo, Vicuna-33B, and Vicuna-13B, with only a marginal difference to human performance, indicating a strong potential in understanding and implementing continuous coordination tasks.

## 5.2 Sustained Coordination and Robustness to Partners

Sustained coordination refers to the ability of agents to continuously collaborate and adapt their actions over extended periods. Robustness to Partners is about an agent's ability to adjust and adapt to interacting with new or unseen partners. We use GPT-4 as the LLM of choice to test these aspects. The choice of LLM is dictated by the fact that only GPT-4 is able to display satisfactory reasoning that will be required consistently for Sustained Coordination. We evaluate the LLM-Co agent on 400 timesteps of gameplay in the Overcooked-AI environment. The evaluation metric used in Overcooked Carroll et al. (2019a) is the sparse reward obtained when one whole delivery is completed by the agents. Each delivery wins the agents 20 points.

We use Self Play with Proximal Policy Optimization (PPO) and Population-Based Training with PPO as the baselines for comparing AI-AI gameplay. For benchmarking AI-Human Proxy gameplay, we use the method of using a **PPO agent trained with a human model** (Behavior Cloning model trained on Human-Human gameplay data) established in Carroll et al. (2019a) and observed in the follow-up works Li et al. (2023); Zhao et al. (2023); Lou et al. (2023); Yu et al. (2023) approaching Zero Shot Coordination.

| | Layouts | | | | |
|---|---|---|---|---|---|
| Agent Type | Cramped Rm. | Asymm. Adv. | Coord. Ring | Forced Coord. | Counter Circ. |
| $PPO_{SP}$ | $198.8 \pm 4.06$ | $167.2 \pm 3.63$ | $\mathbf{190.8} \pm 4.25$ | $151.9 \pm 3.28$ | $122.3 \pm 3.80$ |
| PBT | $216.9 \pm 1.31$ | $190.1 \pm 8.64$ | $173.8 \pm 18.27$ | $169.5 \pm 10.09$ | $140.1 \pm 13.86$ |
| LLM-Co | $\mathbf{220} \pm 0$ | $\mathbf{280} \pm 0$ | $180 \pm 0$ | $\mathbf{200} \pm 0$ | $\mathbf{160} \pm 0$ |

Table 1: Comparison of game play between self-play baselines (PPO, and PBT) and LLM-Co Agents. LLM-Co agents outperform RL methods on 4 out of 5 layouts, demonstrating highly effective reasoning under sustained coordination.

**The LLM-Co agent efficiently completes the Overcooked-AI challenge over a long horizon** We pair two LLM-Co agents together to jointly coordinate and complete the cooking and delivery task in Overcooked-AI. This is analogous to testing agents trained with self-play methods being asked to jointly perform the task. We observe through visualizations of the gameplay that LLM-Co agents make effective use of all resources available to them to complete multiple deliveries effectively. In fact, without being trained or fine-tuned for the task, LLM-Co agents outperform or nearly match Self-Play baselines trained using Proximal Policy Optimization Schulman et al. (2017) or Population-Based Training Jaderberg et al. (2017) which are the gold standard for Multi-Agent Tasks on. Table 1 shows a numerical summary of the scores obtained by agents. These scores represent averages obtained from 100 runs across with standard deviation across 5 seeds for MARL agents. For LLM-Co agents, the score obtained by agents for a fixed game description and directives remains the same as the agent always chooses to take the same medium-level action for a given state and history. This outcome is noteworthy because it demonstrates that Language Learning Models (LLMs), specifically GPT-4 OpenAI (2023) in this case, can outperform RL agents at cooperative multi-agent tasks with minimal scaffolding. We observed that LLM Agents are capable of achieving sustained coordination, adjusting to their partners, and correcting their own actions consistently.

**The LLM-Co agent is robust to the choice of partner.** It is highly likely that an agent is paired up with a biased or sub-optimal partner. It is known that self-play agents, when paired with humans, tend to struggle because their behavior diverges from what they consider to be the optimal strategy

| Agents | Layouts | | | | |
|---|---|---|---|---|---|
| | Cramped Rm. | Asymm. Adv. | Coord. Ring | Forced Coord. | Counter Circ. |
| BC | $103.5 \pm 3.38$ | $136.5 \pm 7.00$ | $59.0 \pm 5.38$ | $20.5 \pm 4.33$ | $38.0 \pm 3.99$ |
| $PPO_{BC}$ | $156.4 \pm 1.48$ | $72.6 \pm 19.44$ | $126.4 \pm 3.24$ | $58.9 \pm 2.98$ | $69.5 \pm 2.18$ |
| LLM-Co | $\mathbf{160} \pm 0$ | $\mathbf{180} \pm 0$ | $\mathbf{160} \pm 0$ | $\mathbf{120} \pm 0$ | $\mathbf{140} \pm 0$ |
| **Playing from swapped positions:** | | | | | |
| BC | $110.0 \pm 3.39$ | $137.5 \pm 8.40$ | $70.0 \pm 4.00$ | $31.0 \pm 5.00$ | $44.0 \pm 3.02$ |
| $PPO_{BC}$ | $163.9 \pm 1.61$ | $\mathbf{178.8 \pm 2.65}$ | $129.8 \pm 3.59$ | $76.9 \pm 2.29$ | $57.6 \pm 2.50$ |
| LLM-Co | $\mathbf{180} \pm 0$ | $140 \pm 0$ | $\mathbf{160} \pm 0$ | $\mathbf{80} \pm 0$ | $\mathbf{120} \pm 0$ |

Table 2: Comparison of AI-Human Proxy Game play. We compare Behavior Cloning Agents, PPO_BC Agents with LLM-Co agents utilizing the GPT-4 LLM. The LLM-Co agents are able to outperform or match the performance of Reinforcement Learning models, indicating that LLM agents are robust to the choice of partner agents.

Carroll et al. (2019a). The LLM-Co agent, on the other hand, does not face this issue. Their actions are based on verbal reasoning, and they adapt to the current situation rather than adhering to a determined policy. Consequently, they outperform Self-play-based methods trained with human data at AI-human proxy gameplay as shown in table 5.2.

**The LLM-Co agent generates explainable outputs through free-text**  Reinforcement Learning (RL-based) agents lack the ability to provide an **underlying rationale** for their actions, making it challenging to understand how their actions contribute to broader objectives. This understanding is crucial for the development of safer and more reliable agents, as well as for debugging when unexpected behaviors occur. The LLM-Co agent addresses this gap by generating medium-level actions and providing high-level reasoning for such a selection. This allows us to extract comprehensive insights into the decision-making processes under given conditions by examining the "analysis" generated by the language model during gameplay. Using insights derived from this explainability, we report qualitative case studies in Appendix B

## 5.3   EXPLICIT ASSISTANCE

| Conditions | Layouts | |
|---|---|---|
| | Locked | Gated Delivery |
| Without Helper Directive | 160 | 0 |
| With Helper Directive | 240 | 180 |

Table 3: Comparison of Gameplay in the Overcooked-Assistance Layouts with and without Helper Directive. The results indicate that the Large Language Model needs to be prompted to be aware of situations where their partner might need assistance in order to be effective in the Overcooked-Assistance layouts.

Finally, we test the ability of the LLM-Co agent to provide explicit assistance to their partners in the new Overcooked layouts defined in 3.4, where proactive help is necessary to complete the task.

**the LLM-Co agent requires a helping directive to choose to help**  The LLM-Co agent, provided with the same prompt and directives as used in Overcooked-AI, struggles to recognize and help their partner agents in Locked and Gated Delivery environments. However, a simple directive informing the LLM-Co agent to "help their partners when the situation demands" makes them actively look for opportunities to help their partners. We see that agents tend to help partner agents during the time they are waiting for their own soup to be cooked by choosing the open gates for the waiting agent. While this is not the most efficient strategy, which would have been to always help out a waiting agent, it still points to the agent's ability to explicitly help out during coordination. Table 5.3 shows

scores obtained by agents over 400 time steps. Both Locked and Gated Delivery require agents to notice a partner in need and consequently benefit from adding a directive to help out their partner.

| | Layouts | |
|---|---|---|
| Agents | Locked | Gated Delivery |
| $PPO_{SP}$ | $132.83 \pm 7.31$ | $134.88 \pm 5.99$ |
| PBT | $175.8 \pm 1.69$ | $178.6 \pm 9.76$ |
| LLM-Co | $\mathbf{220 \pm 0}$ | $\mathbf{180 \pm 0}$ |

Table 4: Comparison of Gameplay on Overcooked-Assistance Layouts between RL baselines and LLM Agents. The RL baselines being able to effectively solve the deliveries indicates that the environments are solvable through self-play training. The high scores achieved by LLM agents demonstrate that LLM agents are capable of reasoning for providing explicit assistance to their partners.

**The LLM-Co agent outperforms MARL methods at Overcooked-Co-op layouts**  Table 5.3 shows the performance of Self Play agents trained using PPO and PBT and compares it with the abilities of the LLM-Co agent provided with a helper directive. Since a reward is provided to both agents for delivery in Self Play training, we expected to see them gain the ability to open gates as it results in a reward after a short delay. In spite of this, the LLM-Co agent has the upper hand in their ability to deduce the right actions required to facilitate their partners.

## 6 CONCLUSION

In this study, we evaluate the reasoning abilities of Large Language Models for achieving Multi-agent coordination. We evaluate LLMS across five aspects necessary for coordination through comprehensive evaluations in three different environments. We introduce The LLM-Co Framework for enabling Large Language Models to play multi-agent coordination games. We also curate the LLM-ToM-Reasoning dataset to assess the Theory of Mind inference and Situated Reasoning Abilities of Large Language Models. We show that LLM Agents are capable of Sustained Coordination in the Overcoked Environment and are Robust to the choice of partner. Finally, we introduce two new layouts to the Overcooked-AI environment and demonstrate the ability of Large Language Models to provide explicit assistance to their partners during coordination games.

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

# A  DETAILS FOR USING LLM-CO FRAMEWORK WITH OVERCOOKED

## A.1  GAME AND LAYOUT DESCRIPTION

We use a general game description $G$ that explains the rules and objectives of overcooked. Since each layout has a different number of locations, like onion dispensers and cookers, we include a succinct description of each environment $L_i$, which includes how many instances of particular facilities there are. For environments that include partitions, we mention which partition each of the agents is situated in and what facilities that agents can access. In addition, we also mentioned the shape of the environment.

## A.2  STATE REPRESENTATION

The State Representation Module programmatically converts the state information into a natural language description $D(S)$, which can be processed by a Large Language Model (LLM). The state $S$ includes variables that fully represent the necessary details of the layout as well as the players. The information provided in $D(S)$ is equivalent to what would be accessible to a Reinforcement Learning (RL) agent in the form of state representations. We refer to the Blue agent as Alice and the Green agent as Bob. The following information is included in $D(S)$:

**Objects Held by Each Player**   The state description $D(S)$ begins by detailing the inventories $I_{\alpha_1}$ and $I_{\alpha_2}$ of Alice and Bob, respectively. Each inventory $I_{\alpha_i}$ (where $i \in \{1, 2\}$) can contain one of the following items: {"onion", "plate", "cooked soup"}. This inventory information is translated into natural language and incorporated into $D(S)$ in the format: "You are holding $I_{\alpha_1}$. Bob is holding $I_{\alpha_2}$." Such information is vital for inferring the likely subsequent actions of the partner agent.

**Location of the Agent Controlled by LLM:**   Given the limitations of Large Language Models (LLMs) in interpreting grid-based spatial information, we opt to provide processed location data to the LLM. For each agent $P_i$ (where $i \in \{1, 2\}$), and for each location of interest denoted as loc, we calculate the distance $d_{(P_i, \text{loc})}$ as the number of steps required to reach loc from $P_i$ using the shortest available path. The state description $D(S)$ then includes this processed location information in the format: "loc is $d_{(P_i, \text{loc})}$ units away." Here, loc can represent various points of interest such as onion dispensers, plate dispensers, cookers, delivery areas, kitchen counters, or shared counters. If a location is either inaccessible or blocked by another agent, this is explicitly stated in $D(S)$. For example, if a location is blocked by Bob, it would be stated as "loc is blocked by Bob." To distinguish between the location information relevant to each agent, $D(S)$ prefixes the respective sections with "Your location information:" for the agent controlled by the LLM and "Bob's location information:" for the partner agent.

**Cooker Information**   The state description $D(S)$ also incorporates information about the cooker, which is central to the gameplay strategy. Specifically, for each cooker $i$, $D(S)$ includes the number of onions $n_i$ currently in the pot. Additionally, $D(S)$ provides the operational state of the cooker, denoted as $\text{CookerState}_i$, which can be either "Off" or "On". Lastly, the current condition of the soup in the cooker is represented by $\text{SoupState}_i$, which can take one of the following values: "Cooking", "Cooked", or "Not Started". Thus, the information for cooker $c_i$ is formatted as: "$c_i$ has $n_i$ onions. $c_i$ is $\text{CookerState}_i$. Soup in $c_i$ is $\text{SoupState}_i$."

**Kitchen Counter Information**   The state description $D(S)$ includes information about kitchen counters, which are primarily used for temporary object storage. Specifically, $D(S)$ identifies the closest empty kitchen counter $k_{\text{empty}}$ and the set $K_{\text{filled}}$ of all counters currently holding an object.

**Shared Counter Information**   Shared counters serve as specialized kitchen counters for object transfer between agents. For each shared counter $i$, $D(S)$ includes the status for $s_i$, as "$s_0$ is empty" or "$s_1$ contains onion," to offer a complete environmental overview. Unlike kitchen counters, where only the closest empty counter is mentioned, all empty shared counters are mentioned.

Table 5: Complete high level action set.

| High-level Actions | Description |
|---|---|
| pick up onion from oX. | Pick up an onion from onion dispenser number X. |
| pick up plate from pX. | Pick up a plate from plate dispenser number X. |
| put onion in cX. | Place the onion into cooker number X. |
| put soup on plate from cX. | Serve the soup from cooker number X onto a plate. |
| deliver soup in dX. | Deliver the soup to delivery location number X. |
| pick up onion from sX. | Pick up an onion from shared counter number X. |
| pick up plate from sX. | Pick up a plate from shared counter number X. |
| place onion on sX. | Place the onion on shared counter number X. |
| place plate on sX. | Place the plate on shared counter number X. |
| pick up onion from kX. | Pick up an onion from kitchen counter number X. |
| pick up plate from kX. | Pick up a plate from kitchen counter number X. |
| place onion on kX. | Place the onion on kitchen counter number X. |
| place plate on kX. | Place the plate on kitchen counter number X. |
| wait. | wait for one time-step. |
| move away. | Randomly move away from the current location away from the other agent. |

## A.3 FEASIBLE ACTION GENERATION:

Table 5 shows our full high-level action set used for the Overcooked environment. These are directives that can be selected by the LLM based on the provided state information. The action set is inspired by the planning objectives used by Carroll et al. (2019a) in their coupled planning game play. The action set is complete in the sense that an agent can utilize the action set to complete multiple deliveries in the Overcooked AI environment.

The constrained action set generator verifies the feasibility of performing an action before making it available to the LLM. This is a rule-based program that reduces the set of available actions. For example, It is not possible to pick up an onion or a plate if the agent is already holding an onion. In such a case, we will remove actions like *"pick up onion from o0"*, *"pick up plate from p0"* from the set of available actions, which will be provided to the LLM along with the state description at each turn.

## A.4 THE LARGE LANGUAGE MODEL

The LLM takes the game description, environment description, state description, and the feasible action set as its input, along with a history of the previous actions (5 actions.) It then selects an action from the set of feasible actions and formats its response as *Analysis: ⟨analysis⟩. Action:⟨action⟩.* The LLM is asked to elucidate the current situation, including the environment state, a guess about the other player's intention (ToM), and the explanation behind their next action in the analysis section.

## A.5 ACTION MANAGER

This module is tasked with converting the high-level actions chosen by the Large Language Model (LLM) into specific, executable steps. Upon receiving a directive from the LLM, the module uses a Breadth-First Search algorithm to identify the shortest path to the target. The immediate next step along this path is then selected as the agent's action for that moment.

The module maintains control until it fully executes a complex directive, such as *"place onion in c0"*. This directive would include both the sequence of movements needed to reach the target location and the final action to complete the task, like placing the onion. Once this directive is completed, control returns to the State Representation Module, which then consults the LLM for the next high-level action.

The Action Manager also handles stalemate situations where both agents contend for the same spot or each other's spots using a combination of querying the LLM for another action and deterministic move-away action.

This design approach relieves the LLM from the complexities of low-level motion planning, an area where it typically struggles. It also reduces the number of calls to the LLM, saving both time and computational resources.

# B    CASE STUDIES

**LLMCo Agents are capable of long-term planning while considering their partner's intentions.** The Overcooked-AI environment, and cooperative tasks in general, require agents to plan ahead while considering their partner agents' intentions. Figure 6 provides a visual summary of the LLM-Co agent's rationale behind selecting the action "pick up a plate from p0" when there are already two onions in the pot with the partner agent having already picked up the third one. The LLM-Co agent reasons that their partner is probably going to place their onion in the cooker next and consequently decides to take the preemptive action of picking up a plate.

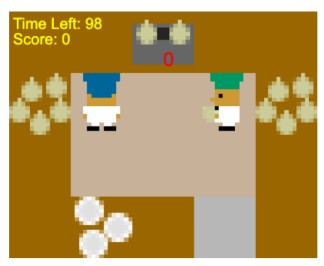

**Explanation**:Since Bob is already holding an onion, you should pick up a plate for the soup that is about to be cooked

**Action**: pick up plate from p0.

Figure 6: The LLM-Co agent (Blue) understands that its partner has already picked up an onion and will probably end up placing it in the cooker to complete the requirement. So, it chooses to "pick up a plate" which would help in the next stage.

**LLM-Co Agents enable their partners.** Effective coordination requires agents to have an understanding of how they might be inhibiting their partners. Figure 7 shows the LLM-Co agent's reasoning in a situation where two cookers are located in a corner (Coordination Ring environment), and one of them is cooking the soup while the other is missing some onions. Since the soup is not cooked, the agent with a plate is just waiting near the cooker to load the soup onto the plate. However, this time could be used by the other agent to put an onion in the adjoining cooker. In this situation, the LLM-Co agent is able to realize that it needs to move away to enable its partner.

**Explanation**:You already have a plate with soup in your inventory, and c0 is still cooking. Bob is holding an onion and can add it to c1. Your best action is to deliver the soup you have to maximize deliveries. Bob's next action should be to add the onion to c1 and start cooking.

**Action**: move away.

Figure 7: The LLM-Co Agent (Blue) thinks about the partner's mind states (shown in red) and makes an informed decision (highlighted in green) to move away to make way for the partner agent)

**LLM-Co Agents are sensitive to their partner's actions and correct their behaviors based on the resulting state changes.** In the Overcooked-AI environment, each agent gets to perform one action at every time step. This leads to situations where an agent might end up taking an action that seems optimal in the situation but is soon realized to be sub-optimal due to their partner's action. A situation like this is demonstrated in Figure 8. The Blue agent has picked up an onion but needs a plate. In such a situation we observe that the agent is able to realize its sub-optimal action selection and notice that a plate has been made available by the other agent. In this case, the LLM-Co agent places the object on the nearest available counter, freeing its hands. In the subsequent step, it goes and picks up a plate to load the soup.

**Explanation**: Both cookers are already full and cooking, and holding onto the onion will prevent you from performing other actions. Placing the onion on the nearest counter will free up your hands for the next task.

**Action**: place onion on k5.

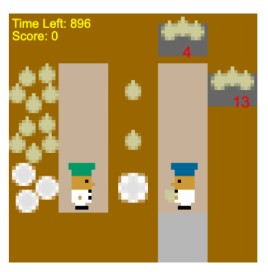

Figure 8: The LLM-Co Agent (Blue) realizes that the onion it picked up cannot be used and decides to correct itself by placing it on the kitchen counter to free up its hands.

**Explanation**: Since the soup in c0 is still cooking and Bob is already holding a cooked soup but can't deliver it because d0 is inaccessible to him, the best action would be to open g0 so Bob can deliver his soup and you can continue cooking.

**Action**: open g0.

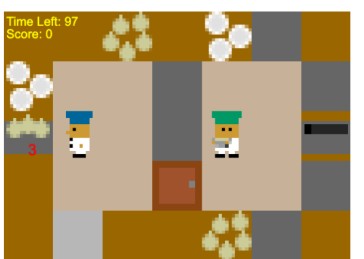

Figure 9:

**LLM-Co agents explicitly help out their partners by opening gates** Figure 9 shows a visual summary of the reasoning followed by LLM-Co agents before choosing to open gates. It can be clearly seen that they realize the imminent rise in efficiency that will result from allowing their partners to access the delivery areas and recognize that the right way to do this is by opening the gate for them.

## C  Prompt Details for Overcooked

**Overcooked Task Description Prompt:**

```
In the game Overcooked, I am Alice, my teammate is Bob.
LAYOUT_DESCRIPTION.
We must coordinate to make onion soups with 3 onions each.
Once a soup is cooked it needs to be placed on a plate and
delivered.  I can only carry one item at a time.  My goal is to
maximize the number of deliveries.  I want to be efficient and
prepare for the next soup while the current soup is cooking.  I'll
provide my action history, current state, teammate's status, and
my possible actions.  Help me select the best action from the
list.  Format your response as:  Explanation:<Brief explanation
for next action including a prediction of Bob's next action>.
Action:  <action>.  Only select one action.  Do not say anything
else.  Got it?
```

**Overcooked Coordination Task Description Prompt:**

```
In the game Overcooked, I am Alice, my teammate is Bob.
LAYOUT_DESCRIPTION.
We must coordinate to make onion soups with 3 onions each.
Once a soup is cooked it needs to be placed on a plate and
delivered.  I can only carry one item at a time.  My goal is to
maximize the number of deliveries.  I want to be efficient and
prepare for the next soup while the current soup is cooking.  I'll
provide my action history, current state, teammate's status, and
my possible actions.  I want to prefer helping the other player
with their cooking and delivery if the situation arises.  Help
me select the best action from the list.  Format your response
as:  Explanation:<Brief explanation for next action including a
prediction of Bob's next action>.  Action:  <action>.  Only select
one action.  Do not say anything else.  Got it?
```

**Layout Prompts (Overcooked):**

---

`Cramped Room`

---

The environment is rectangular with 2 onions dispensers (o0, o1), cooker (c0), plate dispenser (p0) and delivery area (d0). Additionally there are kitchen counters (k0 to k8) which can be used to temporarily store onions and plates while you do something else. Objects on counters can be picked up later and should be considered as they may be closer than items in dispensers.

---

`Asymmetric Advantages`

---

There are two partitions in the current environment. Bob is in the left partition with access to onion dispenser (o0), delivery area (d0), plate dispenser (p0) and kitchen counters (k0, k1, k2, k3, k4, k11, k12, k16, k18, k20, k21, k22, k23). Alice is in the right partition and has access to onion dispenser (o1), delivery area (d1), plate dispenser (p1) and kitchen counters (k6, k7, k8, k9, k10, k14, k15, k17, k19, k25, k26, k27, k28). Both have access to both cookers (c0, c1). Kitchen counters (k0 to k28) can be used to temporarily store onions and plates while you do something else. Objects on counters can be picked up later and should be considered as they may be closer than items in dispensers.

---

`Forced Coordination`

---

The environment is split into two partitions, one with each player. In the right partition, Alice has access to cookers (c0, c1), delivery area (d0) and kitchen counters (k6, k8, k12). In the left partition, Bob has access to onion dispensers (o0, o1), plate dispenser (p0) and kitchen counters (k1, k10). Kitchen counters can be used to temporarily store onions and plates while you do something else. Both players have access to shared counters (s0, s1, s2) which can be used to transfer onions and plates depending on the situation. Note that the objects on the shared counters can be accessed by both players. Objects on counters can be picked up later and should be considered as they may be closer than items in dispensers.

---

`Coordination Ring`

---

The environment is narrow and circular, with onion dispensers (o0, o1), plate dispenser (p0), cookers (c0, c1), and a delivery area (d0). Additionally there are kitchen counters (k0 to k10) which can be used to temporarily store onions and plates while you do something else. Objects on counters can be picked up later and should be considered as they may be closer than items in dispensers.

---

`Counter Circuit`

---

The environment is circular with two onion dispensers (o0, o1), plate dispenser (p0), cookers (c0, c1) and delivery area (d0). There are also the shared counters (s0, s1, s2, s3) which can be used to pass objects from one player to the other. Additionally there are kitchen counters (k0 to k15) which can be used to temporarily store onions and plates while you do something else. Objects on counters can be picked up later and should be considered as they may be closer than items in dispensers.

---

**Layout Prompts (Overcooked Assistance):**

```
Gated Delivery
────────────────────────────────────────────────────
The environment is rectangular with 2 onions dispensers (o0, o1),
cooker (c0) and plate dispenser (p0).  The delivery area (d0) is
inaccessible behind closed gates and can be accessed by opening
one of the gates (g0, g1).  A gate can only be opened by a player
if they are not carrying an object.  Once the gate is opened it
will only stay open for a brief time and then close on its own.
```

```
Locked
────────────────────────────────────────────────────
The environment is divided into 2 partitions.  Alice is in
the left partition with access to onion dispenser o0, plate
dispenser p1, cooker c0, and delivery area d0.  Bob is in the
right partition with access to onion dispenser o1, plate dispenser
p0, and cooker c1.  The two partitions are connected by gate g0
which can be opened if a player is holding nothing.  Opening the
gate will allow players to move freely between partitions.  The
gate will close after enough time automatically.
```

