# OpenReview forum: "Evaluating Multi-Agent Coordination Abilities in Large Language Models"
_ICLR.cc/2024/Conference — ICLR 2024 Conference Withdrawn Submission_

### Official Review · Reviewer_3Qgp · 2023-10-29

**Soundness:** 2 fair
**Presentation:** 2 fair
**Contribution:** 2 fair
**Rating:** 5
**Confidence:** 4

**Summary:**

The paper looks at the problem of using LLMs as a way to facilitate coordination in multi-agent settings in general and humans specifically. The paper starts by trying to evaluate the ability of current LLM systems in mental modeling and situated reasoning. Based on the results, they focus on GPT-4 (which performed best) and build a coordination framework around it. The framework itself is reasonably straightforward, with LLMs more or less being provided the current state and game description and then expected to come up with the right course of action. There are some minimal wrappers on action extraction and management. The central hypothesis seems to be that powerful LLM models can be used for coordination processes out of the box. They show how their proposed LLM-Co framework outperforms RL agents in multiple multi-agent collaborative benchmark problems along various metrics.

**Strengths:**

The paper puts forth a rather direct hypothesis on the usability of LLMs as a decision-making system that can support coordination. While the paper doesn’t exploit it, the fact that LLM supports full natural language could also make it a very attractive mechanism to allow AI-human collaboration. The results do show the current system works much better than many of the current state-of-the-art RL methods.

**Weaknesses:**

Unfortunately, I have multiple concerns with regard to the current papers. First off, for the evaluation of TOM capabilities, 18 scenarios seem far too small a set to make definitive claims about TOM and situated reasoning capabilities. However, we can put this aside for now, given this is not the central claim being evaluated, and as the paper points out, there are other works like (Kosinski 2023) that have tried to provide a more detailed evaluation of at the very least the TOM capabilities. This experiment mostly seems to be a way for the authors to zero in on GPT-4 as the core LLM to focus on.

My next concern is the related works that are being overlooked. The biggest one is the work done on solving no-press diplomacy (particularly [1] and [2]). This is a particularly important work since it represents state-of-the-art coordination methods for MARL settings and actually uses large language models as part of its coordination mechanism, especially in [2]. Your work should be compared and contrasted against these earlier works. Another set of works that are overlooked is the works showing limitations in reasoning and planning capabilities (examples include [3] and [4]). This second set of works, for at the current moment, implies a real upper threshold on the kind of problems that can be solved using your methods—something the RL methods don’t necessarily face.

Finally, it seems to me that the LLM agent and the RL agents seem to be operating over action space with different levels of abstraction. At the very least, the kind of actions mentioned in the examples, like “pickup plate from p0”, seems to be at a higher level of temporal abstraction than what is usually used by RL agents trained on overcooked domains. If this is in fact, the case it is not clear how that would be a fair comparison. For example, there might be a large amount of task-specific reasoning going on in the action manager that might avoid many minor collaboration issues.


Minor Comments:
Now for some minor comments:
The use of cites is a bit distracting. When you are not directly referring to the authors please use \citep.
Also, it is not clear how a post hoc generated rationale by the LLM is supposed to help. Is this information meant to be used by the user, or was it just used in the paper for illustrative purposes? If it’s the former, doesn’t it run into issues related to hallucination and the system potentially providing unreliable information?

[1] Bakhtin, Anton, et al. "Mastering the game of no-press Diplomacy via human-regularized reinforcement learning and planning." arXiv preprint arXiv:2210.05492 (2022).

[2] Meta Fundamental AI Research Diplomacy Team (FAIR)†, et al. "Human-level play in the game of Diplomacy by combining language models with strategic reasoning." Science 378.6624 (2022): 1067-1074.

[3] Srivastava, Aarohi, et al. "Beyond the imitation game: Quantifying and extrapolating the capabilities of language models." arXiv preprint arXiv:2206.04615 (2022).

[4] Valmeekam, Karthik, et al. "On the Planning Abilities of Large Language Models--A Critical Investigation." arXiv preprint arXiv:2305.15771 (2023).

**Questions:**

Please answer the questions related to the related work and the actions used by the RL agents.

---

### Official Review · Reviewer_cHGX · 2023-10-31

**Soundness:** 3 good
**Presentation:** 2 fair
**Contribution:** 2 fair
**Rating:** 3
**Confidence:** 4

**Summary:**

This paper proposes the LLM-Coordination Framework to build agents with LLMs to play cooperative games. They consider five aspects of the agent's ability and evaluate them in three games to show the potential of LLM-based agents to achieve multi-agent coordination.

**Strengths:**

1. Good coverage of evaluation aspects: five aspects are proposed to evaluate the coordination ability of LLM agents.

2. Good reproducibility: the source code is provided and it is easy to reproduce the result.

**Weaknesses:**

1. Novelty is unclear: prior to this work, many other frameworks to build cooperative LLM agents have been proposed like [1, 2, 3]. The difference from existing frameworks and the novelty of the proposed framework is unclear.

2. Lack of discussion on the proposed evaluation aspects: the five evaluation aspects are reasonable in the overcooked environment, but are they sufficient or necessary to evaluate the multi-agent coordination ability of LLM in (most) coordination tasks?

3. Insufficient evaluation:

    1. Lack of other environments: the ToM and Reasoning aspects are evaluated in three environments, other four aspects are only evaluated in the overcooked environment. Also, the CollbaCapture and CollabEscape are not common in the literature. It is better to add results on other common environments.

    2. Lack of baselines: LLM-Co is only compared with learning-based methods like PPO and PBT, but not with other LLM agents. [1] also builds LLM agents for the overcooked game. The authors should at least include this baseline and compare with it.

    3. Lack of human experiment: the best way to evaluate AI-Human coordination is to let real humans play with the agent, but this work only uses a BC policy and a PPO_BR policy. The human experiment is a standard evaluation part in prior works on overcook like [4, 5, 6].

Reference:
[1] Zhang, Ceyao, et al. "Proagent: Building proactive cooperative ai with large language models." arXiv preprint arXiv:2308.11339 (2023).

[2] Zhang, Hongxin, et al. "Building cooperative embodied agents modularly with large language models." arXiv preprint arXiv:2307.02485 (2023).

[3] Chen, Weize, et al. Agentverse: Facilitating multi-agent collaboration and exploring emergent behaviors in agents. arXiv preprint arXiv:2308.10848, (2023).

[4] Strouse, D. J., et al. "Collaborating with humans without human data." Advances in Neural Information Processing Systems 34 (2021): 14502-14515.

[5] Zhao, Rui, et al. "Maximum entropy population-based training for zero-shot human-ai coordination." Proceedings of the AAAI Conference on Artificial Intelligence. Vol. 37. No. 5. 2023.

[6] Yu, Chao, et al. "Learning Zero-Shot Cooperation with Humans, Assuming Humans Are Biased." arXiv preprint arXiv:2302.01605 (2023).

**Questions:**

1. What is the novelty and difference of LLM-Co from existing frameworks to build cooperative agents?

2. What is the specific design in LLM-Co to enable coordination? What is the ablation result when the design is removed?

3. Are the five aspects sufficient or necessary to evaluate the multi-agent coordination ability of LLM in (most) coordination tasks?

4. Evaluation results in other common environments.

5. Evaluation results of other frameworks like [1, 2, 3].

6. Human experiment result.

---

### Official Review · Reviewer_GAhq · 2023-11-03

**Soundness:** 2 fair
**Presentation:** 3 good
**Contribution:** 2 fair
**Rating:** 3
**Confidence:** 4

**Summary:**

The paper explores the multi-agent coordination capabilities of LLMs. It proposes a new framework and three games designed to test the Theory of Mind, Situated Reasoning, and sustained coordination skills. It finds that among the LLMs tested,  GPT-4 achieved the best performance and outperformed RL methods. Also, LLM power agents can explain their decision-making process in natural language. The authors claim that the results suggest that LLMs have the potential for complex, interactive tasks requiring coordination with humans or other AI agents.

**Strengths:**

Overall, I enjoyed reading the paper. The ideas presented are interesting. The authors tested relevant concepts for effective teaming, such as ToM reasoning and backup behaviours. Three games evaluate some of these aspects, and they used Overcooked to test actual team performance. These present reasonable early choices to demonstrate using the LLM-powered agents as potential collaborators, and the results are promising against RL baselines.

**Weaknesses:**

While I enjoyed the paper and found the ideas interesting, the paper lacks sufficient discussion of the choices that authors may have made to reach the outcomes. Naturally, the prompt design is important; there needs to be more discussion of the decisions taken when designing the prompts.

The discussion of the results could be improved. For example, what could we learn from the scenarios where the agents did not do well at optimal action reasoning (Fig 5)? In Fig 5, the final column is how humans performed -- who were these participants, and how many?

Finally, the task performance was tested in one task, Overcooked, with two agents. This restricts the generalizability of the ideas.

I have further questions, which I outline below.

**Questions:**

1) Did the authors face any size restrictions when designing the prompts? If yes, how were these solved?

2) How did the authors decide the contents of the prompt, what information to include, and what to exclude?

3) How did the LLM Co Agents perform in the individual games, i.e., were there any interesting observations that were made in a specific game? Fig 5 reports the aggregate performance. It would be helpful to include a breakdown by game type in the Appendix.

4) Where results involve human performance, who were these participants, and how many? How did you recruit?

5) What could we learn from the scenarios where the agents did not do well at optimal action reasoning (Fig 5)?

6) How would you expect the performance to be impacted by increasing team size?

**Details Of Ethics Concerns:**

I did not find sufficient discussion on who the human subjects were (Fig 5).